# Probiotics Supplementation Attenuates Inflammation and Oxidative Stress Induced by Chronic Sleep Restriction

**DOI:** 10.3390/nu15061518

**Published:** 2023-03-21

**Authors:** Yadong Zheng, Luyan Zhang, Laura Bonfili, Luisa de Vivo, Anna Maria Eleuteri, Michele Bellesi

**Affiliations:** 1School of Biosciences and Veterinary Medicine, University of Camerino, 62032 Camerino, MC, Italy; 2Center for Neuroscience, University of Camerino, 62032 Camerino, MC, Italy; 3School of Food and Biological Engineering, Zhengzhou University of Light Industry, Zhengzhou 450001, China; 4School of Pharmacy, University of Camerino, 62032 Camerino, MC, Italy; 5School of Physiology, Pharmacology and Neuroscience, University of Bristol, Bristol BS8 1TD, UK

**Keywords:** probiotics, sleep deprivation, inflammation, oxidative stress, microglia

## Abstract

**Background:** Insufficient sleep is a serious public health problem in modern society. It leads to increased risk of chronic diseases, and it has been frequently associated with cellular oxidative damage and widespread low-grade inflammation. Probiotics have been attracting increasing interest recently for their antioxidant and anti-inflammatory properties. Here, we tested the ability of probiotics to contrast oxidative stress and inflammation induced by sleep loss. **Methods:** We administered a multi-strain probiotic formulation (SLAB51) or water to normal sleeping mice and to mice exposed to 7 days of chronic sleep restriction (CSR). We quantified protein, lipid, and DNA oxidation as well as levels of gut–brain axis hormones and pro and anti-inflammatory cytokines in the brain and plasma. Furthermore, we carried out an evaluation of microglia morphology and density in the mouse cerebral cortex. **Results:** We found that CSR induced oxidative stress and inflammation and altered gut–brain axis hormones. SLAB51 oral administration boosted the antioxidant capacity of the brain, thus limiting the oxidative damage provoked by loss of sleep. Moreover, it positively regulated gut–brain axis hormones and reduced peripheral and brain inflammation induced by CSR. **Conclusions:** Probiotic supplementation can be a possible strategy to counteract oxidative stress and inflammation promoted by sleep loss.

## 1. Introduction

Sleep is a fundamental behavior that fills approximately one-third of a human’s lifetime and is critical for both physical and mental well-being [1]. Chronic sleep restriction (CSR), defined as insufficient/inadequate sleep over a prolonged period of time, is prevalent in contemporary society owing to professional obligations and lifestyle habits [2,3]. Epidemiological investigations have estimated that about 30% of adults and adolescents regularly experience insufficient sleep [4]. CSR can lead to a range of brain deficits, including impaired attention and learning, and is associated with increased risk of neuropsychiatric disorders, but also cardiovascular diseases and metabolic alterations [4,5,6,7]. Growing evidence has demonstrated that CSR is linked to a low-grade inflammation, as reflected by increased inflammatory plasma cytokines and by the presence of other markers of inflammation in the brain, such as activation of microglia cells [8,9,10]. In addition, insufficient sleep can lead to the accumulation of intracellular reactive oxygen species (ROS) and/or reactive nitrogen species (RNS), resulting in an unbalance between the oxidant and antioxidant systems of the body [10,11]. Excessive ROS and RNS can react with carbohydrates, proteins, lipids, and DNA, and therefore, causes oxidative stress-related cellular damage and increased risk of disease, and in extreme cases, even death [11,12]. Sleep deprivation also affects energy homeostasis and has been associated with perturbed blood levels of peptide hormones, including ghrelin, leptin, and glucagon like peptide 1 (GLP-1) [13,14].

Probiotics have been attracting increasing interest in recent years for their ability to ameliorate inflammation-related illness. Numerous studies suggested that probiotics can effectively reduce both peripheral and central inflammation through multiple pathways. The underlying mechanism is associated with rebalancing of gut flora alteration, improvement of gut permeability, and modulation of immune function with lower production of proinflammatory cytokines [15,16,17]. Furthermore, probiotics can regulate microglia maturation and activity, and may also prevent neuroinflammatory processes, with positive impact in a series of diseases, such as inflammatory bowel disease, obesity, and neurodegenerative conditions [18,19,20]. Furthermore, it has been observed that probiotics and/or bacterial metabolites can interact with the host by modulating the level of both endogenous and exogenous ROS, ultimately improving oxidative status [21,22,23]. Long-term supplementation with multi-strain probiotic formulation exerted antioxidant and neuroprotective effects in a transgenic Alzheimer’s disease mouse model by activating the silencing information regulator 2 related enzyme 1 (SIRT1) pathway [24].

Several studies have provided evidence that sleep deprivation can perturb the composition of gut microbiota [25,26]. By inducing a breakdown of the intestinal epithelial barrier, sleep disruption may favor the passage of bacteria and their end-products, thus affecting the host and promoting immune reaction and inflammation [27]. Thus, sleep loss-associated inflammation may depend, at least in part, on an alteration of the gut microbiota physiology. There is also evidence that administration of probiotics can improve sleep. Manipulation of the gut microbiota through the administration of single or multi-strain probiotics can ameliorate sleep quality by reducing the Pittsburgh Sleep Quality Index (PSQI), a common indicator reflecting the impairment of sleep quality [28,29].

Here, we tested the hypothesis that chronic oral supplementation with a multi-strain probiotic formulation can reduce oxidative stress and inflammation induced by CSR. To this end, we administered a mixture of several probiotic strains (SLAB51) or vehicle in normal sleeping mice and in mice exposed to CSR, and we assessed the extent of oxidative damage and inflammation in the brain and at systemic level using biochemical and morphological methods.

## 2. Material and Methods

### 2.1. Materials

SLAB51 was provided by Ormendes SA (Jouxtens-Mezery, Switzerland, https://agimixx.net (accessed on 23 February 2023). SLAB51 is a multi-strain probiotic formulation that contains eight different live bacterial strains: *Streptococcus thermophilus* DSM 32245, *Bifidobacterium lactis* DSM 32246, *Bifidobacterium lactis* DSM 32247, *Lactobacillus acidophilus* DSM 32241, *Lactobacillus helveticus* DSM 32242, *Lactobacillus paracasei* DSM 32243, *Lactobacillus plantarum* DSM 32244, and *Lactobacillus brevis* DSM 27961. Polyvinylidene difluoride (PVDF) membranes and reagents for western blotting analyses, and the oxyblot protein oxidation protein detection kit for carbonyl groups introduced into proteins were obtained from Merck KGaA (Darmstadt, Germany). All antibodies used for western blotting, including nitrotyrosine, dityrosine, 4-hydroxynonenal (4-HNE), 8-oxoguanine DNA Glycosylase (OGG1), 8-oxo-2′-deoxyguanosine (8oxodG), ionized calcium-binding adapter molecule 1 (IBA-1), Interleukin 6 (IL-6), Interleukin 10 (IL-10), tumor necrosis factor alpha (TNFα), Interleukin 1 beta (IL-1β), and glyceraldehyde 3-phosphate dehydrogenase (GAPDH) antibody, were purchased from AbCam (Milan, Italy). Anti-IBA-1 for immunohistochemistry was purchased from Wako (019-19741), while fluorescent secondary antibodies were purchased from Thermo Fisher (Monza, Italy). ELISA Kits for IL-1β, TNF-α, IL-10, and IL-6 cytokines determination in plasma were obtained from Thermo Fisher Scientific Inc. (Italy). ELISA Kits for ghrelin, leptin, and GLP-1 measurement in plasma were from Merk-Millipore (Milan, Italy). Proteases inhibitors tosyl phenylalanyl chloromethyl ketone (TPCK) were from Merck KGaA (Darmstadt, Germany). Proteins immobilized on films were detected with the enhanced chemiluminescence (ECL) system (Amersham Pharmacia Biotech, Milan, Italy). A list of all the antibodies and their concentrations is given in Appendix A.

### 2.2. Animals

Eight-week-old wild-type male B6128SF2 (n = 28, weight 25–35 g) mice were acquired from the Jackson Laboratory (Bar Harbor, ME, USA). All mice were housed in groups of four in environmentally controlled cages for the duration of the experiment (12 h light/dark cycle, light on at 8:00 P.M., the temperature of 24 ± 1 °C; food and water available ad libitum and replaced daily at 9:00 A.M.). The mouse body weight was measured before and after experimental conditions. All the experiments were performed according to the local Institutional Animal Care and Use Committee and the European Communities Council Directives (2010/63/EU). All appropriate measures were taken to minimize pain and discomfort in experimental animals.

### 2.3. Experimental Design and SLAB51 Administration

Mice were divided into two weight-balanced groups, the water group (w) and the probiotic (p) group. Both the water and the probiotic groups were further separated into chronic sleep restriction (CSR-w, n = 7, CSR-p, n = 7) and normal sleep groups (S-w, n = 7, S-p, n = 7). CSR-p and S-p were administered with SLAB-51 dissolved in the drinking water, while CSR-w and S-w received only water. After 8 weeks of treatment, mice started the CSR experiment, during which the probiotic group was still fed with probiotics until the end of the experiment (Figure 1).

The dosage of SLAB51 (200 billion bacteria/kg/day) was calculated using the body surface area principle based on our previous experiments. Before starting the experiment, we estimated the daily water intake and dissolved the proper amount of probiotics into the drinking water to reach the desired concentration [30]. We have previously checked the viability and stability of the probiotic formulation after dissolution in water at 21 ± 5 °C. Fluorescence microscopy was used to measure the proportion of vital bacteria, which indicated that 88 percent of the strains survived after 30 h under the aforementioned conditions [24]. The fresh drinking solution was changed every day. The body weights of mice were monitored every 2 weeks before treatment and subsequently weekly during the experiment to ensure normal experimental food consumption.

### 2.4. CSR Procedure

CSR was achieved by an automated sleep deprivation chamber (Pinnacle Technology inc.). The effectiveness of this automated sleep deprivation method has been proved in previous experiments using EEG recording in rodents [31,32]. The procedure consists of a slow rotating bar placed at a short distance above the cage floor, lightly nudging the animal from sleep and encouraging low levels of activity until the animal maintains wakefulness on its own. Mice were sleep restricted for 7 consecutive days. To ensure a modest but persistent sleep restriction, mice were exposed to the rotating bar for 24 h/day at a velocity of 2 rpm (one turn each 30 s). Control mice were placed in the same sleep deprivation chambers and allowed to sleep undisturbed, except for 3 h/day (during the dark period, when mice are usually awake) during which the bar rotation was activated to expose the mice of this group to the experience of the bar movement and to the stress associated with it. Mice were housed in groups of four with ad libitum access to food and water or drinking bottle containing SLAB51. Animal behavior was daily assessed by direct visual observation. After 7 days, all mice were sacrificed between 9:00 and 11:00 A.M. to maintain the time of tissue collection within the same 2-h time of day window for all experimental groups.

### 2.5. Tissue Collection

Mice were anesthetized with isoflurane (1–1.5% volume) and sacrificed by cervical dislocation. Brains were extracted—one hemisphere was processed for biochemical assessments, while the other one was immersed in cold 4% paraformaldehyde dissolved in 0.1M phosphate buffer for fixation. Blood samples were collected from the abdominal aorta with a heparinized syringe connected to a 26 G needle and collected in EDTA tubes. They were centrifuged at 3500 rpm for 10 min at 4 °C, and the obtained plasma was promptly supplemented with Pefabloc 1 mM for subsequent cytokine ELISA detection.

### 2.6. Western Blotting Analyses

Mouse brain tissue was homogenized in a solution of 50 mM Tris buffer, 150 mM KCl, 2 mM EDTA, and pH 7.5 (1:5 weight/volume of buffer). Brain homogenates were immediately centrifuged at 13,000× *g* for 20 min at 4 °C, and the supernatants were collected upon adding proteinase inhibitors (1 mM tosyl phenylalanyl chloromethyl ketone (TPCK) and Pefabloc). Protein concentration was determined using the Bradford protein assay [33]. Brain homogenates were analyzed through western blotting to investigate the following protein expression levels: 3-Nitrotyrsosine (3-NT), Dityrosine, 4-HNE, OGG1, 8-oxodG, and IBA-1. In detail, brain samples (30 μg total protein) were loaded on 10–12% sodium dodecyl-sulphate-polyacrylamide gel electrophoresis (SDS-PAGE) and transferred onto polyvinylidene fluoride (PVDF) membranes. Following incubation with the specific antibodies, the immunoblot detection was performed with an enhanced chemiluminescence (ECL) western blotting ChemiDocTM System (Biorad, Milan, Italy). Molecular weight markers (6.5 to 205 kDa) were included in each gel. Glyceraldehyde-3-phosphate dehydrogenase (GAPDH) was used to ensure equal protein loading and to normalize western blot data. ChemiDoc acquired images or scanned autoradiographs (16 bit) were processed through ImageJ software (NIH) to calculate the background mean value and its standard deviation. The background intensity mean was then subtracted from the raw digital data to obtain a background-free image. For each band, the integrated densitometric value was determined as the sum of the density values for all pixels belonging to the analyzed band with a density value larger than the background standard deviation. The ratios of band intensities were calculated within the same western blot. All the calculations were carried out using Matlab (The MathWorks Inc., Natick, MA, USA).

### 2.7. Oxyblot Analysis

The Oxyblot protein oxidation protein detection kit was used to determine protein carbonyl groups. According to the manufacturer’s instructions, brain homogenates (15 μg total proteins) were incubated at room temperature with 2,4-dinitrophenylhydrazine (DNPH) to generate 2,4-dinitrophenylhydrazone (DNP-hydrazone). The DNPH-derivatized samples were subsequently separated by SDS-PAGE and electroblotted onto the PVDF membrane. Then, the membrane was incubated with an anti-DNP antibody followed by a specific secondary antibody. The ECL system was utilized for the detection. To examine the same protein load, prior to incubation with an anti-DNP primary antibody, a reversible Ponceau stain was applied. The statistical significance was determined by comparing the densitometric values of oxyblot bands (oxidation level) to those stained with Ponceau red (protein content).

### 2.8. Plasma Cytokines Levels

The levels of inflammatory cytokines IL-1β, TNF-α IL-6, and IL-10 in plasma were measured using an enzyme-linked immunosorbent assay NOVEX^®^ ELISA kit (Invitrogen^TM^, Waltham, MA, USA), according to the manufacturer’s instructions and detected using a SpectraMax ABS Plus microplate reader (Molecular Devices, Germany).

### 2.9. Ghrelin, Leptin, and GLP-1 Determination

Concentrations of gut–brain axis hormones were measured through ELISA in mouse plasma treated with protease inhibitors (Pefabloc and TPCK). We used a sandwich ELISA based on the capture of ghrelin, leptin, or GLP-1 (active form) in the plasma by specific monoclonal IgG. After the binding of a second biotinylated antibody to ghrelin, leptin, or GLP-1, the unbound material was washed. The remain complex was conjugated to horseradish peroxidase and the quantification of immobilized antibody-enzyme conjugates was performed by monitoring horseradish peroxidase activities in the presence of the substrate 3,3,5,5-tetra-methylbenzidine. The enzyme activity was measured spectrophotometrically by the increased absorbance at 450 nm, corrected from the absorbance at 590 nm, after acidification of formed products using a SpectraMax ABS Plus microplate reader (Molecular Devices, Germany).

### 2.10. Immunohistochemistry

Brain tissue was allowed to fix for 10 days at 4 °C and then was cut on a vibratome in 50 μm coronal sections. Sections were rinsed in a blocking solution [3% bovine serum albumin (BSA) and 0.3% Triton X-100] for 1 h and incubated overnight (4 °C) in the same blocking solution containing anti-IBA-1 (1:500). Sections were then probed with secondary antibodies: anti-rabbit Alexa Fluor 594 (1:600)-conjugated secondary antibodies. Sections were examined with a confocal microscope (Nikon Eclipse Ti, Tokyo, Japan). For IBA-1, microscopic fields (n = 5 per section, 1 section per mouse) were randomly acquired as 1024 × 1024-pixel images (pixel size, 561 nm; Z-step, 750 nm) in mouse frontal cortex using a UPlan FL N 40x objective (numerical aperture, 1.3). To improve the signal/noise ratio, two frames of each image were averaged.

Image analysis. For IBA-1 staining, all analyses were performed on maximum-intensity projections (Z-project, Maximum Intensity function in ImageJ) of the 21 images constituting the Z-stack. Individual microglial cells were counted and manually segmented using the Single Neurite Tracing plug-in of FIJI [34]. Microglia process arborization was quantified using Sholl analysis by measuring the number of intersections between microglial branches and each Sholl ring.

### 2.11. Statistical Analysis

Statistical analysis was performed using Graphpad prism software (La Jolla, CA, USA) and Matlab (The MathWorks Inc., Natick, MA, USA). One-way ANOVA was used for western blotting, ELISA experiments, and microglia density, while two-way ANOVA was used for microglial morphological branching analysis where the between-subject factor were the groups, and the within-subject factor were the Sholl rings. One or two-way ANOVA was followed by the Tukey post-hoc test. Alpha was set to 0.05 and appropriately corrected for multiple comparisons.

## 3. Results

### 3.1. Probiotics Administration Ameliorates CSR-Induced Protein and Lipid Oxidation

To verify the antioxidant effect of SLAB51, we measured the levels of protein and lipid oxidation by quantifying carbonyls, nitrotyrosine, dityrosine, and 4-HNE in the brain homogenates of all groups using western blotting. We found that levels of carbonyl groups, nitrotyrosine, dityrosine, and 4-HNE considerably increased in the CSR-w relative to the S-w group (*p* = 0.003, *p* = 0.0092, *p* = 0.0054, and *p* = 0.0118, respectively), confirming the reported role of CSR in promoting oxidative stress. SLAB51 administration reduced the CSR-induced effects on oxidative stress as the levels of nitrotyrosine, dityrosine, and 4-HNE were significantly lower in CSR-p than CSR-w (*p* = 0.0052, *p* = 0.0217, and *p* = 0.032, respectively) and no longer significantly different between CSR-p and S-p (*p* > 0.05), with the only exception of carbonyl levels that showed only a trend (CSR-p vs. CSR-w, *p* = 0.0666, Figure 2).

### 3.2. Probiotics Treatment Improves DNA Antioxidant Capacity

In order to determine the effect of SLAB51 on DNA oxidation, the expression of the DNA base excision repair enzyme OGG1 and the DNA oxidation product 8-oxodG were measured in brain homogenates of all groups of mice. In the water group mice, CSR had no significant effect on OGG1 levels compared to control, whereas the probiotic treatment significantly increased OGG1 levels in both S-p (*p* = 0.029) and CSR-p mice (*p* = 0.0043), confirming the antioxidant effect of SLAB51 treatment. By contrast, we found higher levels of 8-oxodG in CSR-w mice compared to S-w (increased by 27 ± 9.4%; *p* = 0.049). This difference was no longer present in CSR mice treated with SLAB51 (CSR-p vs. S-w, *p* = 0.85; Figure 3).

### 3.3. Probiotics Reduces CSR-Induced Neuroinflammation and Systemic Inflammation

We probed the effects of SLAB51 on neuroinflammation by measuring the expression level of several markers of inflammation, including IL-1β, TNF-α, IL-6, and IL-10 cytokines. In brain homogenates, CSR-w mice showed significantly enhanced expression of TNFα (*p* = 0.011) and IL-1β (*p* = 0.048), and attenuated expression of IL-6 (by 56 ± 9.8%, *p* = 0.023) relative to S-w. By contrast, SLAB51 treatment reduced the expression of IL-1β and TNF-α (CSR-p vs. CSR-w, IL-1β: *p* = 0.024; TNF-α: *p* = 0.0013), while it increased the expression of IL-6 (CSR-p vs. CSR-w, *p* = 0.014) and IL-10 (CSR-p vs. CSR-w, *p* = 0.017). We also measured the levels of the microglia-specific expression marker IBA-1 in brain homogenates. We found that CSR increased the expression of IBA-1 by 28 ± 8.9% (S-w vs. CSR-w, *p* = 0.040) in the water group; this effect was blunted by probiotic administration (CSR-p vs. CSR-w, *p* = 0.013, Figure 4). At the systemic level, SLAB51 administration had no effect on plasma cytokines concentrations in S-w mice. Similar to the brain compartment, CSR significantly increased the plasma concentration of TNF-α (*p* = 0.005) and IL-1β (*p* = 0.015), while it decreased the concentrations of IL-6 (*p* = 0.002) and IL-10 (*p* = 0.001). Long-term SLAB51 supplementation effectively restored changes in the plasma cytokines levels in CSR mice (S-w vs. CSR-w, TNF-α: *p* = 0.0031; IL-1β: *p* = 0.0023; IL-6: *p* = 0.016; IL-10: *p* = 0.0089, Figure 5).

### 3.4. Probiotics Attenuate Morphological Microglial Changes Promoted by Sleep Loss

Low-grade neuroinflammation can be associated with morphological changes of microglia, such as process retraction [35]. Hence, we quantified microglia process arborization using Sholl analysis (Figure 6). We manually segmented 2108 microglial cells in the frontal cortex (S-w, n = 606; S-p, n = 520; CSR-w, n = 383; CSR-p, n = 599) and found an effect of condition (F (3, 1289) = 4.355; *p* = 0.0046). Specifically, the administration of probiotics did not promote morphological changes in microglial cells in the normal sleeping mice (S-w vs. S-p, *p* = 0.4025; Figure 7A). When comparing CSR-w with S-w, we found a significant difference in the level of process arborization between these groups, with CSR-w mice showing a net decrease in the number of processes relative to S-w (*p* = 0.0032, Figure 7B). By contrast, the mice exposed to CSR that received probiotics showed comparable levels of process arborization to those of normal sleeping mice (CSR-p vs. S-p, *p* = 0.1359, Figure 7C) and significantly more than CSR-w (*p* = 0.0489; Figure 7D). Furthermore, we quantified microglia density, finding no changes in the density of cells in all groups of mice (F (3, 24) = 0.7390; *p* = 0.55; Figure 8). These results indicate that probiotic administration can attenuate the microglia morphological changes induced by sleep loss.

### 3.5. Probiotics Restored Blood Concentrations of Gut–Brain Axis Hormones

To confirm the role of gut–brain axis modulation by probiotics, we measured the blood concentrations of key hormones involved in the gut–brain axis. Concentration of ghrelin, leptin, and GLP-1 were importantly modified by CSR. Specifically, we found an increase of ghrelin (*p* = 0.0025) and GLP-1 (*p* = 0.0068) and a decrease of leptin (*p* = 0.0004) in CRS-w relative to S-w. These changes were dampened by SLAB1 supplementation, with ghrelin, leptin, and GLP-1 concentration levels being comparable to those of the sleeping mice and statistically different from those of CSR-w mice (Ghrelin: *p* = 0.047; Leptin: *p* = 0.0045; GLP-1: *p* = 0.04, Table 1).

## 4. Discussion

In this study, we found that a week of CSR induced oxidative stress in the mouse brain, promoted brain and systemic inflammation, and altered the gut–brain axis. The oral administration of SLAB51 boosted the antioxidant capacity of the brain, thus limiting the oxidative damage provoked by loss of sleep. Moreover, it restored the levels of gut–brain axis hormones and attenuated the development of peripheral and brain inflammation induced by CSR.

Sleep is critical for maintaining body and brain functions, and insufficient sleep has been related to increased risk for a variety of diseases (e.g., type 2 diabetes, obesity, neurodegeneration, depression, anxiety, etc.), for which oxidative damage and inflammation have been proposed as potential underlying mechanisms [36,37,38].

Oxidative stress is the result of an unbalance between the production of reactive oxygen species (ROS) or reactive nitrogen species (RNS) and the antioxidant capacity of the cells [39]. Proteins, lipids, and DNA are major targets of ROS or RNS in biological systems [40,41]. There is no general consensus on the role of sleep loss in promoting oxidative stress [42]. Some studies found that sleep loss can either increase the production of oxidative radicals or lower antioxidant responses, while others did not find any change in oxidative stress markers or antioxidant capacity in peripheral blood or brain regions following sleep loss [43,44,45,46,47]. This discrepancy has been ascribed to the different sleep deprivation procedures (e.g., gentle handling vs. disk over the water) and the different duration of the sleep deprivation. While acute sleep loss appears to up-regulate the antioxidant cellular machinery, chronic loss of sleep weakens the antioxidant response, thus suggesting that extended wakefulness may be more likely associated with oxidative stress [38]. In our study, we found that 7 days of chronic sleep restriction were capable of increasing the brain levels of carbonyls, nitrotyrosines, and dityrosines, all well-established markers of protein oxidation. In parallel, we found augmented levels of 4-Hydroxynonenal (HNE), a major end product that is derived from the oxidation of lipids. Furthermore, studies on people over age 60 have shown that even one night of sleep loss can induce the expression of genes involved in DNA damage and aging [48]. Animal studies have also shown that sleep deprivation can cause genetic damage in a variety of organs [49]. The accumulation of DNA damage has been linked to DNA mutations, altered gene expression in the brain, and cognitive decline [50]. In our study, we observed increased DNA oxidation in CSR mice compared to control animals, as detected by the decreased OGG1 expression and the increased 8-oxodG levels in the brain homogenates. We also found increased plasma levels of ghrelin in the CSR-w group. Besides its role in regulating appetite, ghrelin has been recently proposed as a systemic oxidative stress sensor [51]. Collectively, these data suggest that chronic sleep loss can lead to oxidative damage of proteins, lipids, and DNA.

Sleep loss has also been repeatedly associated with heightened inflammation. Increased plasma levels of numerous cytokines including IL-1, TNF-a, IL-6, IL-17, nuclear factor-kappa B (NFkB), and altered numbers and activity of macrophages and natural killer cells have been found after both acute and chronic sleep deprivation in healthy individuals [52,53,54]. These findings were also supported by preclinical studies, in which the inflammatory markers IL-1, IL-6, and TNF-a were found elevated in the peripheral blood and several brain regions [55,56]. While the mechanisms through which sleep loss leads to an inflamed state are unclear, oxidative stress may contribute to inflammation by stimulating the release of proinflammatory cytokines, including TNF-α and IL-1β [38]. Furthermore, ROS and RNS can activate other inflammatory mediators such as NF-kB and vascular cell adhesion molecule-1 (VCAM-1) [57]. In this study, CSR was associated with an increase of TNF-α and IL-1β and a decrease of IL-6 and IL-10 in the brain and peripheral blood. While IL-10 is a well-recognized anti-inflammatory cytokine, the role of IL-6 in sleep deprivation-related inflammation remains unclear. Some studies showed that total sleep deprivation or sleep fragmentation led to increases in plasma IL-6 levels, which were interpreted as a marker of inflammation [58,59]. However, other works demonstrated that IL-6 has a crucial anti-inflammatory role in local and systemic inflammatory responses by modulating levels of proinflammatory cytokines [60,61]. Our previous study showed that CSR could activate microglia without affecting the levels of cytokines in the cerebral spinal fluid [8]. Sustained microglia activation could potentially increase the brain’s vulnerability to various types of damage [8]. In this study, we confirmed the activation of microglia induced by CSR, but we also found an increase of IBA-1 expression in the brain of CSR mice. IBA-1 has been demonstrated to have a role in actin-crosslinking of microglial membrane ruffling, and its expression relates to the microglial activation since membrane ruffling is required for the shift from quiescent ramified to activated amoeboid microglia [62]. All together, these findings confirm that sleep loss is associated with increased systemic and brain inflammation.

Probiotics are live microorganisms intended to change the composition of the flora of the gastrointestinal tract of the host and provide health benefits when consumed [63]. The mechanisms through which probiotics improve health are numerous and include the modulation of the host immune system, modification of the intestinal microbiota, protection against physiological stress, pathogen antagonisms, and improvement of the barrier function of the gut epithelium [64]. In this study, we used SLAB51, a multi-strain probiotic supplementation. Previous research has shown that SLAB51 could restore normal eubiosis in animal models of neurodegenerative disorders [30,65], while it had little/no effect on the microbiota of healthy wild-type mice [30]. The effects of probiotics on inflammation and oxidative stress biomarkers have been extensively investigated in animal models and clinical trials. Most recent studies confirmed the role of probiotics in decreasing the levels of CRP, high-sensitivity(hs)-CRP, and TNF-a levels [66,67,68,69]. Previous work using SLAB51 found that this formulation increased the relative abundance of gut anti-inflammatory bacteria such as *Bifidobacterium* spp. and decreased the concentrations of pro-inflammatory *Campylobacterales*, consequently regulating inflammatory pathways. Moreover, it promoted the proliferation of bacteria that produced short-chain fatty acids (SCFAs). It is, thus, possible that the enriched gut concentration of anti-inflammatory and neuroprotective SCFAs contributed to reduce the plasma levels of pro-inflammatory cytokines and to enhance the concentrations of anti-inflammatory cytokines [30]. Consistent with these results, a recent meta-analysis evaluating 42 controlled trials demonstrated that levels of several pro-inflammatory cytokines (i.e., IL-12, IL-4, etc.) were significantly lowered by probiotic supplementation [68]. By contrast, levels of IL-10, glutathione, nitric oxide, total antioxidant status, and total antioxidant capacity were significantly increased with probiotics administration [66,67,70,71,72]. It is worth noting, however, that several studies have reported no effect of the probiotic treatment in regulating inflammation or the oxidative status relative to the placebo group [68,70,73,74].

The anti-inflammatory effects of probiotics are not limited to the gut. Indeed, chronic low-grade inflammatory processes are now thought to play an etiological role in the pathogenesis of several neuropsychiatric disorders and probiotics have been proposed as potential compounds capable of mitigating these pathologies by modulating the immune-to-brain signaling and alleviating the chronic immune activation in the brain [75,76]. In that perspective, administration of multi-strain probiotics—including SLAB51—have resulted in reduced neuroinflammation in animal models of Alzheimer’s disease [24,77,78,79]. Moreover, a recent meta-analysis of 5 studies involving 297 subjects has found improved cognitive performance in AD or MCI patients following probiotic supplementation, likely through decreasing inflammatory and oxidative stress levels [80]. In our study, we found that administration of SLAB51 per se affected neither the levels of inflammatory cytokines in the brain and plasma nor the morphology of microglia cells of sleeping animals (S-p similar to S-w), whereas it induced the overexpression of OGG1 in both sleeping and sleep restricted mice (S-p and CSR-p). Thus, SLAB51 supplementation not only did not trigger an inflammatory response per se, but it enhanced cellular antioxidant capacity. More importantly, when administered in mice later exposed to CSR, it contrasted the rise of central and peripheral inflammation and oxidative stress levels, thus indicating that probiotics can abolish the immune and inflammatory response associated with loss of sleep. These findings are consistent with a recent study carried in acutely sleep-deprived monkeys, where supplementation of GABA-producing probiotics reduced the proinflammatory cytokines IL-8 and TNF-alpha, with no effect on the circulating levels of IL-6 and IL-10 [81]. Similarly, supplementation of *Lacticaseibacillus paracasei* was capable of restoring memory deficits in mice subjected to partial sleep deprivation [82]. Although it was not directly tested in this study, it is possible that the effect on cognition was mediated by the immunomodulation properties of probiotics. The mechanism through which probiotics exert their protective role on neuroinflammation is unclear, but it could be attributed, at least in part, to their direct effects on the gut–brain axis [83,84,85]. For example, numerous reports over the past decades have described ghrelin to be a potent anti-inflammatory mediator [86], while high levels of leptin have been related to inflammation [87]. In this regard, we found that probiotic supplementation restored nearly normal plasma concentration of ghrelin, leptin, and GLP-1, whose levels were remarkably altered by CSR. Although indirect, this evidence supports the hypothesis that probiotics can modulate neuroinflammation via the production and release of specific gut hormones [83,84,85,86,87].

In humans, probiotics have been mostly administered to improve sleep quality rather than counteract the effects of sleep loss. A recent systematic review analyzed a total of 14 studies finding that probiotics supplementation significantly reduced Pittsburgh Sleep Quality Index (PSQI) score (i.e., improved sleep quality) relative to baseline, while no significant changes were reported for other subjective sleep quality metrics or objective sleep parameters, such as efficiency and latency. Although not significant, subsequent analysis found that healthy participants had a greater benefit on sleep quality than those with a medical condition and the use of single-strain probiotics was better than multi-strain probiotics in improving sleep quality [88]. The possibility of counteracting the deleterious effects of sleep deprivation with probiotics is intriguing and could be relevant for populations who by necessity have a disrupted sleep schedule, such as shift workers. The few available studies along this direction showed that probiotics have the potential to reduce the magnitude of the stress response that anticipates the beginning of the night shift [89] and alleviate anxiety and fatigue in shift workers [90], but there are still no available data on the role of probiotics in reducing the proven long-term risk in developing shift work-associated diseases.

A number of limitations to the study should be acknowledged. First, we did not use oral gavage to administer probiotics to mice, but we dissolved SLAB1 in the drinking water. This methodological aspect could have generated variability in the intake of probiotics within the probiotic group. However, we treated the animals for an overall 9 weeks (8 weeks before starting the sleep experiment and during the week of the experiment). This long treatment duration ensured adequate intake of probiotics and may have limited heterogeneous effects among animals. Additionally, since we administered probiotics before and during the CSR, we currently cannot distinguish whether probiotics had preventive or therapeutic effects on CSR-induced inflammation and oxidative stress, and future research with a different experimental design is needed to clarify this ambiguity. Furthermore, we did not assess cognitive functions in our mice, and therefore, we do not know whether probiotics were able to restore potential sleep loss-associated cognitive deficits and whether these effects were related to systemic and brain inflammation. Finally, we only studied male mice, and it is possible that probiotics exerted a different influence on inflammation in female mice, although other studies have found that sex had only a minor effect on the immune modulation of probiotics [91].

## 5. Conclusions

Our study provides direct support to the growing evidence that probiotics can attenuate oxidative stress and inflammation in the brain and at systemic level via the gut-brain axis. In addition, it indicates that probiotic supplementation can represent a viable strategy to counteract oxidative stress and inflammation related to sleep loss, thus possibly limiting its negative consequences on health and well-being.

## Figures and Tables

**Figure 1 nutrients-15-01518-f001:**
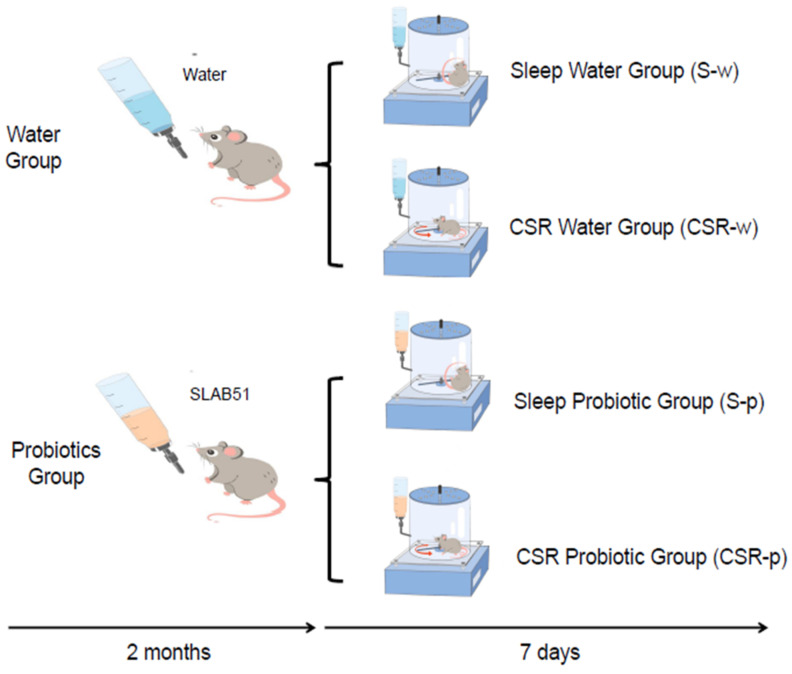
Experimental design.

**Figure 2 nutrients-15-01518-f002:**
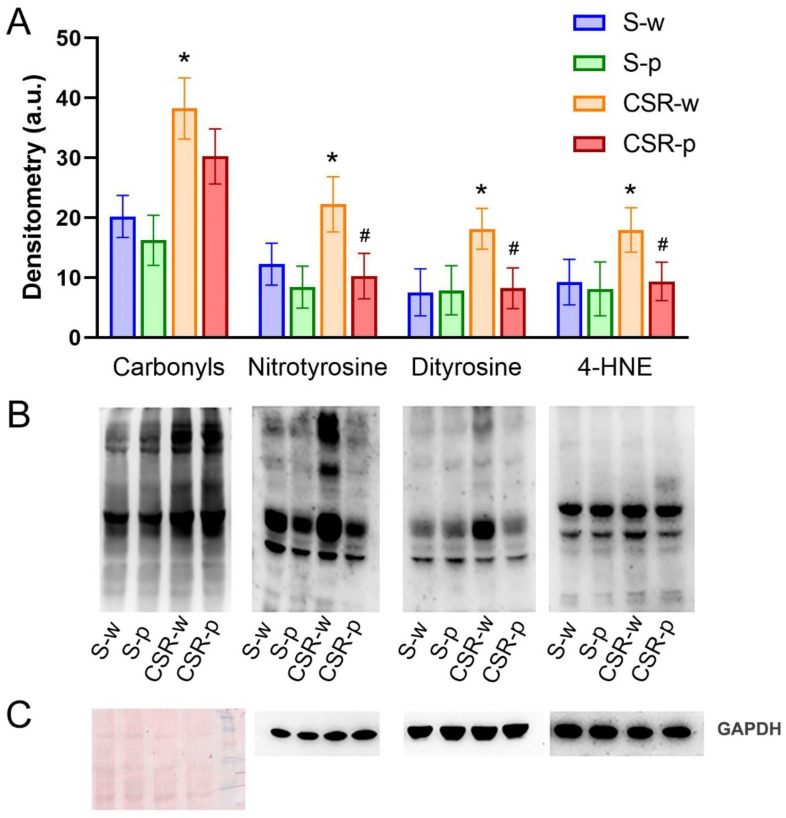
Effects of CSR and SLAB51 on protein and lipid oxidation. (**A**). Quantification of protein carbonyls, nitrotyrosine, dityrosine, and lipid 4-HNE adduct in brain homogenates in all groups of mice. The densitometric analyses obtained from three separate blots. Values are expressed as mean ± standard error. One-way ANOVA results were (F (3, 24) = 9.277; *p* = 0.0003) for carbonyls, (F (3, 24) = 7.550; *p* = 0.001) for nitrotyrosine, (F (3, 24) = 5.431; *p* = 0.00539) for dityrosine, and (F (3, 24) = 3.967; *p* = 0.01183) for lipid 4-HNE. * indicates statistical significance between S-w and CSR-w, while # indicates statistical significance between CSR-w and CSR-p. (**B**). Representative immunoblots for protein carbonyls, nitrotyrosine, dityrosine, and lipid 4-HNE adduct. (**C**)**.** Equal protein loading for nitrotyrosine, dityrosine, and 4-HNE adducts were verified by using an anti-GAPDH antibody. Ponceau staining has been used to check loading in oxyblot. Molecular weight standards (6–205 kDa) were used for molar mass calibration.

**Figure 3 nutrients-15-01518-f003:**
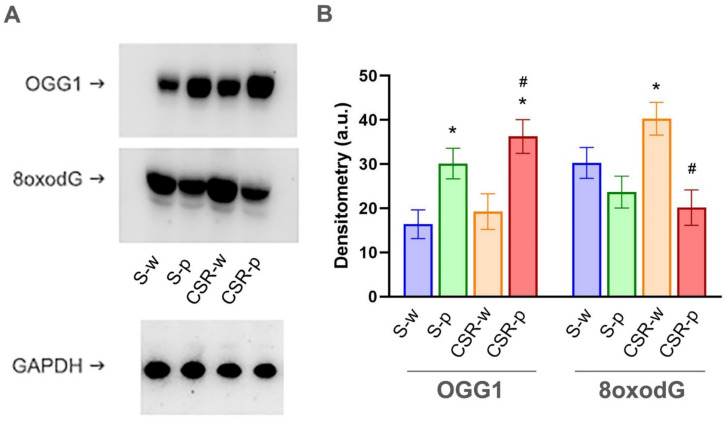
Effects of CSR and SLAB51 on DNA oxidation. (**A**). Representative bands for OGG1 and 8-oxodG levels in brain homogenates of all groups. Equal protein loading was verified by using an anti-GAPDH antibody. (**B**). Quantification of OGG1 and 8-oxodG levels. The densitometric analyses obtained from three separate blots. Values are mean ± SEM. One-way ANOVA results were (F (3, 24) = 6.304; *p* = 0.00262) for OGG1, (F (3, 24) = 6.698; *p* = 0.00192) for 8oxodG. * indicates statistical significance relative to S-w, while # indicates statistical significance between CSR-w and CSR-p.

**Figure 4 nutrients-15-01518-f004:**
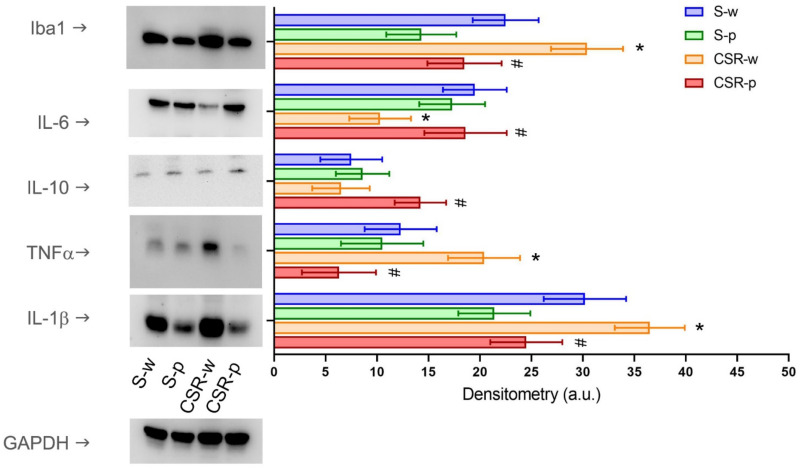
Effects of CSR and SLAB51 on neuroinflammation. Neuroinflammation cytokines levels measured in brain homogenates of all groups. The densitometric analyses obtained from three separate blots and representative immunoblots are shown. Equal protein loading was verified by using an anti-GAPDH antibody. Values are mean ± SEM. One-way ANOVA results were (F (3, 24) = 6.239; *p* = 0.00276) for Iba1, (F (3, 24) = 3.379; *p* = 0.03473) for IL-6, (F (3, 24) = 3.847; *p* = 0.02220) for IL-10, (F (3, 24) = 7.774; *p* = 0.00085) for TNFα, and (F (3, 24) = 4.534; *p* = 0.01179) for IL-1β. * indicates statistical significance between S-w and CSR-w, while # indicates statistical significance between CSR-w and CSR-p.

**Figure 5 nutrients-15-01518-f005:**
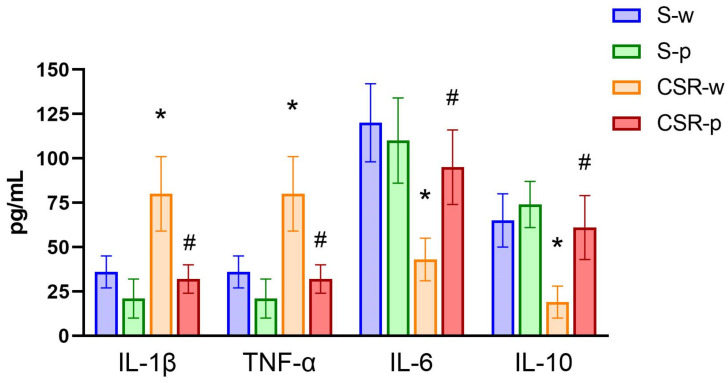
Effects of CSR and SLAB51 on systemic inflammation. Proinflammatory and anti-inflammatory cytokines levels in blood plasma of all groups of mice. Analytes concentrations are expressed as mean ± SEM. One-way ANOVA results were (F (3, 24) = 3.379; *p* = 0.03473) for IL-6, (F (3, 24) = 7.816; *p* = 0.00082) for IL-10, (F (3, 24) = 6.376; *p* = 0.00248) for TNFα, and (F (3, 24) = 9.891; *p* = 0.0002) for IL-1β. * indicates statistical significance between S-w and CSR-w, while # indicates statistical significance between CSR-w and CSR-p.

**Figure 6 nutrients-15-01518-f006:**
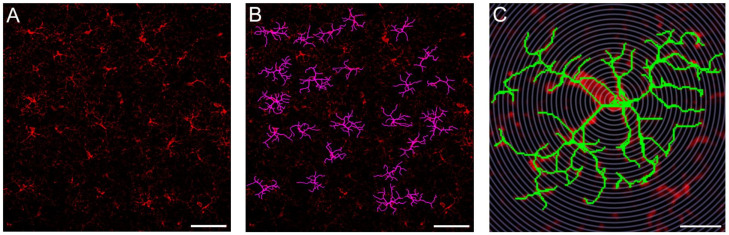
Microglia cells segmentation and Sholl analysis. (**A**) Confocal image of microglia IBA-1 positive cells. Scale bar = 50 μm. (**B**) Examples of manually segmented microglial cells. Scale bar = 50 μm. (**C**) Sholl analysis of a representative microglial cell. Scale bar = 5 μm.

**Figure 7 nutrients-15-01518-f007:**
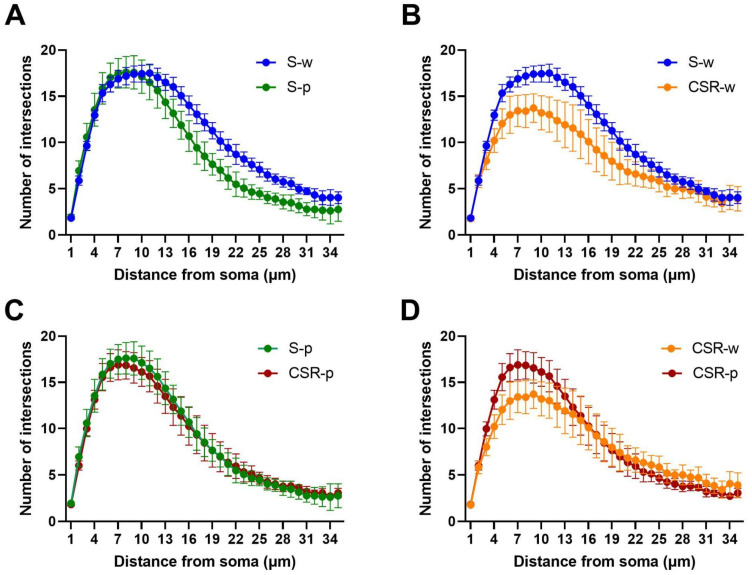
Quantification of the Sholl analysis of microglia process arborization. Number of intersections in S-w and S-p (**A**), in S-w and CSR-w (**B**), in S-p and CSR-p (**C**), and in CSR-w and CSR-p (**D**). N = 7 for each group, values are mean ± SD.

**Figure 8 nutrients-15-01518-f008:**
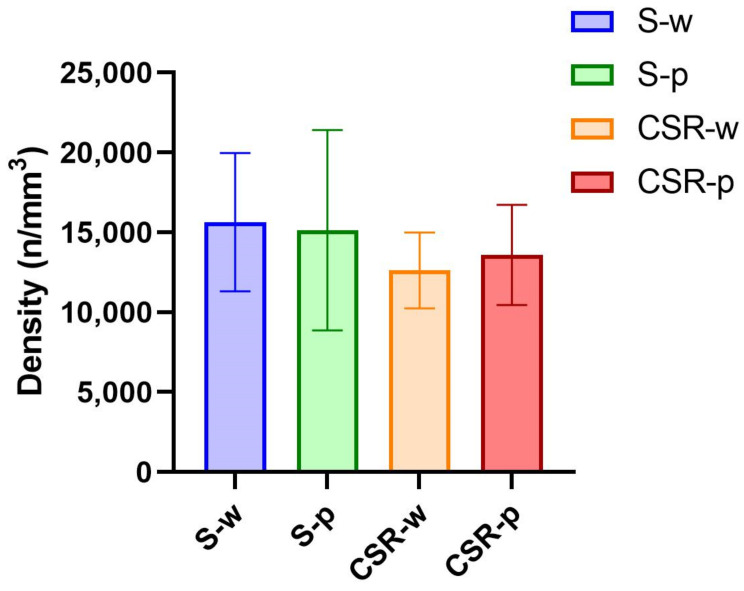
Density of microglia. Quantification of the density of microglial cells in all groups. Values are mean ± SD.

**Table 1 nutrients-15-01518-t001:** Plasmatic levels of ghrelin, leptin, and GLP-1. Values are mean ± SD. One-way ANOVA results were (F (3, 24) = 8.231; *p* = 0.00061) for Ghrelin, (F (3, 24) = 10.952; *p* = 0.00010) for Leptin, and (F (3, 24) = 10.269; *p* = 0.00015) for GLP-1. * indicates statistical significance between S-w and CSR-w, while # indicates statistical significance between CSR-p and CSR-w.

	S-w	S-p	CSR-w	CSR-p
Ghrelin (pg/mL)	458.26 ± 56.45	476.01 ± 46.58	689.37 ± 44.00 *	562.25 ± 42.21 #
Leptin (ng/mL)	11.12 ± 3.40	13.02 ± 2.90	1.99 ± 0.70 *	7.41 ± 3.22 #
GLP-1 (pg/mL)	102.36 ± 32.56	112.23 ± 26.65	602.36 ± 100.23 *	303.26 ± 53.65 #

## Data Availability

The data presented in this study are available on request from the corresponding authors.

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
