# Peer review of "Probiotics Supplementation Attenuates Inflammation and Oxidative Stress Induced by Chronic Sleep Restriction"

_nutrients, 2023, doi:10.3390/nu15061518_

Round 1

Reviewer 1 Report

This study investigated the effect of probiotics supplementation on chronic sleep restriction inflammation and oxidative stress in mice. Overall, the work done is exhaustive, however, as explained below, I raised some questions and suggestions that need to be addressed.

Major points

- Why do the authors claim probiotics have preventive effects and exclude therapeutic effects? Yet the treatment was continued during the week of sleep restriction. In order to differentiate these two distinct effects, two groups of mice need to be added to the experiments. The first group should be treated during the two months preceding the CSR and the second group only receive the treatment during the week of CSR. If this is not possible, the authors should revise the use of the term "preventive" throughout the manuscript and highlight this point in the limits and future direction.

- Was the CSR procedure applied for 3 hours a day? This needs to be mentioned clearly in the CSR procedure section.

- SLAB-51 administration should be further detailed. What is the concentration of these compounds in the water?  

- The statistical analysis section should be more detailed. For instance, it should be indicated for which analysis the one-way ANOVA and for which analysis the two-way was used. Moreover, why the authors used different post hoc analyses (Bonferroni or Tukey)? 

- ANOVA main effect should be added to all figures and table 1.

Minor points

-        Abstract: Line 1: please remove the indentation

-        Discussion, paragraph 3, Line 8: please explain further the differences in sleep deprivation procedures reported in the literature.

Reviewer 2 Report

Article by Zheng et al. is well-written and highlights the importance of probiotic supplementation as a possible strategy to counteract oxidative stress and inflammation promoted by sleep loss. Overall article is scientifically sound, novel, and interesting for researchers. I have a few suggestions and questions

1.     Did authors monitor colonization of these bacteria in the gut? To understand the SLAB51 effect it would be nice to check colonization.

2.     I suggest authors should provide a list of primary and secondary antibody list/table describing companies and sources.

3.     I doubt if IL6 can be described as anti-inflammatory. Literature suggests IL6 is both pro and anti-inflammatory.

4.     Axis title of figure 3 is different than other bar graphs, I suggest making it similar to others.

5.     Do authors have any explanation for the mechanism behind the antioxidative and anti-inflammatory activity of SLAB51?

Round 2

Reviewer 1 Report

I thank the authors for considering my comments and consider that the paper is now in a state of being accepted.